# Nanotopography Evaluation of NiTi Alloy Exposed to Artificial Saliva and Different Mouthwashes

**DOI:** 10.3390/ma15238705

**Published:** 2022-12-06

**Authors:** Zoran Bobić, Sanja Kojić, Goran M. Stojanović, Vladimir Terek, Lazar Kovačević, Pal Terek

**Affiliations:** Faculty of Technical Sciences, University of Novi Sad, Trg Dositeja Obradovića 6, 21000 Novi Sad, Serbia

**Keywords:** biomaterial, nitinol, corrosion, orthodontic archwire, mouthwash, AFM, topography, nano changes, ANOVA

## Abstract

Nitinol (NiTi) alloy is a widely used material for the production of orthodontic archwires. Its corrosion behavior in conditions that exist in the oral cavity still remains a great characterization challenge. The motivation behind this work is to reveal the influence of commercially available mouthwashes on NiTi orthodontic archwires by performing non-electrochemical corrosion tests and quantifying the changes in the nanotopography of commercially available NiTi orthodontic wires. In this study, we examined the behavior of NiTi alloy archwires exposed for 21.5 days to different corrosive media: artificial saliva, Eludril^®^, Aquafresh^®^, and Listerine^®^. The corrosion was characterized by contact mode atomic force microscopy (AFM) before and after the corrosion tests. A novel analysis methodology was developed to obtain insight into locations of material gain or material loss based on standard surface roughness parameters Sa, Sdr, Ssk, and S10z. The developed methodology revealed that fluoride-containing mouthwashes (Aquafresh^®^ and Listerine^®^) dominantly cause material loss, while chloride-containing mouthwash (Eludril^®^) can cause both material loss and material gain. The sample exposed to artificial saliva did not display significant changes in any parameter.

## 1. Introduction

Nitinol (NiTi) is a class of near-equiatomic alloys that consists of roughly equal atomic percentages (at%) of Ti and Ni. Due to its good corrosion resistance and special mechanical properties, NiTi is widely used as an archwire in orthodontic treatment [1]. NiTi owes its good corrosion resistance to a passive film that forms on its surface. Surface film is mostly comprised of titanium dioxide (TiO_2_) and a small amount of nickel oxide (Ni-O). However, low amounts of metallic Ni are also present, which makes the film susceptible to corrosion attack [2,3].

During orthodontic treatment, clinicians recommend to their patients the use of various mouthwashes for maintaining oral hygiene. Mouthwash solutions have various components, and some of them can cause NiTi corrosion. These media could contain fluorine- and/or chlorine-containing compounds which are quite aggressive toward the NiTi alloy and consequently induce its corrosion. Corrosion of NiTi in the oral cavity is undesirable because the products of these processes can react with the surrounding bio environment. Numerous studies reported that the release of Ni-ions in these processes can lead to allergenicity, toxicity, and carcinogenicity in the organism [4,5,6]. Since fluorine- and chlorine-containing compounds in mouthwashes can induce the release of Ni-ions and Ti-ions, these compounds are generally accepted as the primary cause of corrosion and have been a subject of numerous published studies. It was found that fluoride ions degrade the protective titanium oxide films that form on NiTi alloy. The hydrofluoric acid (HF) that forms in these contacts rapidly dissolves Ti and consequently accelerates the alloy corrosion [7,8,9]. In addition, chloride ions have a higher affinity toward metals than oxygen and thus replace it in the protective oxide surface layer. This process leads to depassivation and pitting [10,11,12].

To date, the characterization of the corrosion behavior of NiTi alloys by in vitro experiments has been mostly conducted by employing three approaches. The first approach employed electrochemical tests to assay the electrochemical properties such as the corrosion potential, IR drop, polarization resistance, passive current density, and pitting potential (Eb). These properties were evaluated because they can be linked with alloy biocompatibility [13]. Previous investigations showed that concentrations of fluorine- [7,14,15,16,17,18] and chlorine- [14,19,20] containing compounds in commercially available mouthwashes are sufficient to cause a decrease in the corrosion resistance of NiTi alloy and affect the corrosion of the surface. Two different corrosion mechanisms were detected. Chlorine-containing compounds encourage localized corrosion, while fluorine-containing compounds primarily induce general corrosion of NiTi alloys [14,21].

The second approach is to use atomic absorption spectrometry, or some similar technique, to quantify the presence of Ni and Ti in the medium or their depletion from the surface of the sample. Investigations employing this approach showed that the increased concentration of fluorine- and chlorine-containing compounds leads to an increase in ion release [17,22]. They reported that the concentration of fluorine- and chlorine-containing compounds in commercial mouthwashes causes a detectable amount of Ni and Ti ion release from NiTi alloy [10,23,24]. An important result of investigations presented in [10,13,23,25,26] is that they determined the relation of the Ni-ion release with time. The maximum Ni-ion release occurs during the period from the 8th to the 28th day of exposure, and subsequently notably decreases. Such behavior is linked with the thickening of the surface oxide layer which hampers the ion release from the bulk alloy [3,10,13,23,25,26,27,28,29]. These kinds of experiments provide an advantage in evaluations of the real amount of released material and the characterization of the process toxicity.

The third approach is the characterization of corrosion behavior by analyzing the changes that occurred on the sample surface after the corrosion tests. Standard characterization techniques used in this approach are scanning electron microscopy (SEM) and atomic force microscopy (AFM). These techniques allow detection of nanometric changes on the surface and quantification of these changes by surface roughness parameters. It is reported that both fluorine- and chlorine-containing compounds in media can cause a considerable change of the surface topography [10,26,30,31,32,33] and that the intensity of the corrosion processes can depend on the surface topography [18,23,31,34].

NiTi exposed to commercially available mouthwashes exhibited reduced corrosion resistance and increased release of Ni ions [7,14,15,16,17,18,19,20,30,32,35]. It is reported that commercially available mouthwashes in electrochemical tests caused considerable changes on the NiTi surface [14,16,19,20,21]. However, in non-electrochemical tests changes were not detected [30,31]. Electrochemical measurements by definition apply an external potential/current, and this may alter the system and affect the corrosion rate [36,37,38]. Additionally, one should keep in mind that the methodology employed in the investigations [30,31] does not entirely use the possibility of AFM to detect nanometric changes in surface topography because the analysis was not performed on the same micro locations before and after the corrosion tests. Therefore, this study aims to detect, characterize, and quantify changes in the nanotopography of NiTi orthodontic wires caused by commercially available mouthwashes by utilizing the possibility of AFM to detect changes on the nano level in order to confirm results in previous electrochemical corrosion tests.

## 2. Materials and Methods

The samples used in this study were prepared from one NiTi orthodontic wire (Dentaurum, Ispringen, Germany), with a 0.48 mm × 0.64 mm cross section, in an as-received state. The wire was cut into four equal samples. In order to analyze the surface changes induced by corrosion processes, every sample was marked before the corrosion tests by a small groove made on five locations. These marks served to ensure the sample analysis before and after the corrosion test on exactly the same locations. Before and after each corrosion test and each measurement, samples were cleaned in ultrasonic bath containing 98% ethyl alcohol. Each sample was placed in a plastic 0.5 mL cuvette filled with one out of four corrosive media of interest (saliva or mouthwash), for a duration of 21.5 days at room temperature. The exposure time was chosen to be slightly greater than the average time necessary for reaching the stage of a notable decrease in ion release. During the testing period, the media was not changed. Three kinds of popular commercial mouthwashes were chosen. The investigated samples denotations and corresponding corrosive media used in the tests are presented in Table 1.

Corrosion was characterized through changes in samples’ surface morphology, topography, and chemical composition. For these purposes scanning electron microscopy (SEM), atomic force microscopy (AFM), and X-ray energy dispersive spectroscopy (EDS) were employed. Using these techniques, sample surfaces were analyzed near five marked locations, before and after the corrosion tests. Analyses were carried out on locations without obvious surface defects in the protective oxide layer.

AFM measurements were performed by di CP-II (Veeco, Plainview, NY, USA) device in a contact mode, using a symmetrically etched silicon-nitride tip (BRUKER (Billerica, MA, USA), Model: CONT20A-CP, Part: MPP-31123-10). Scanning parameters were as follows: fast scanning direction X-axis, scanning area 100 × 100 µm, scanning rate 0.5 Hz, setpoint 225 nN, and gain 0.5. Images were acquired with a lateral resolution of 256 × 256 pixels. Although a special procedure for probe-sample (probe-mark) positioning was developed for the employed AFM device, measurements taken before and after the corrosion tests exhibited a slight mismatch. To ensure the exact overlap of analyzed areas, topographic images of 80 × 80 µm and 10 × 10 µm areas were extracted from the initially scanned 100 × 100 µm areas. These extracted images were then used for the evaluation of corrosion effects. Image analysis software SPIP 6.2.0 (Image Metrology, Hovedstaden, Denmark) was employed for the analysis of topographic images and for the calculation of surface roughness parameters.

SEM analyses were performed on the same locations as the AFM using a TM 3030 (Hitachi, Tokyo, Japan) device. Chemical composition was determined by EDS using electron acceleration voltage of 15 keV.

In order to evaluate the trends observed in the obtained quantitative data and evaluate their significance, statistical analyses were employed. One needs to prove that the change in data is not a product of chance but indeed caused by tracked parameters. Standard statistical analysis methods employed for this purpose are one-way analysis of variance (ANOVA) and paired *t* test [39,40,41]. These tests are only suitable for datapoints that follow the standard distribution. Therefore, all values of quantitative parameters were submitted to normality test (Anderson–Darling test, *p* > 0.05) and homogeneity of variance (Levine’s test, *p* > 0.05). Paired *t* test was employed for the evaluation of the changes in surface roughness parameters, on chosen locations, which are induced by a corrosive medium. A paired *t* test was employed to evaluate changes in surface roughness parameters induced by corrosive media. ANOVA and the post hoc Fisher’s least significant difference test, at a confidence level of 95%, were employed for the evaluation of two separate cases. First, for the comparison of the influence of various media on NiTi corrosion, the sort of treatment was chosen as an independent variable and the difference in surface roughness parameters of the same locations was chosen as a dependent variable. Second, for the comparison of initial surface of the samples characterized by surface roughness parameters, the sample and surface roughness parameters were chosen for independent and dependent variables, respectively.

## 3. Results

The EDS analysis revealed that the chemical composition of the sample did not significantly change after exposure to the corrosive media and was comprised of approximately 47 wt%Ti and 53 wt%Ni. The presence of fluorine and chlorine was not detected.

Representative SEM and AFM images of sample surfaces before and after the corrosion test in Eludril CLASSIC^®^ (Sample 3) for the same location are given in Figure 1. Representative images obtained before and after corrosion tests for other mediums can be found in the Appendix A. These images demonstrate the surface morphology and topography of the initial sample surfaces and the changes induced by corrosion processes. The initial surface of the samples is characterized by mostly parallel deep grooves and by much smoother areas between them. These grooves can be up to 3 µm deep. Basic analysis of SEM and AFM 80 × 80 µm area images (Figure 1a–d) did not reveal any noticeable differences induced by the corrosive medium. However, when a comparison is made between the small smooth areas of 10 × 10 µm amidst the grooves (Figure 1e,f), changes in surface nanotopography become observable. It appears that some surface features changed their shape or disappeared after exposure to the corrosive medium.

### 3.1. Surface Roughness Parameters 80 × 80 µm

To confirm observations made by image analysis, a detailed statistical analysis of surface parameters was performed. Average values of arithmetical mean height (Sa) and ten-point height (S10z) parameters, determined on the areas of 80 × 80 µm before and after the tests, are presented in Figure 2. It is noteworthy that the values of the Sa parameter, are on a nanometer scale, while the values of the S10z parameter are on a micrometer scale. This indicates the existence of deep grooves and high peaks on the surfaces of the samples, which agrees with the findings obtained from topographic images. Although all of the samples were produced from the same archwire, differences in the initial surface roughness parameters of approximately 15% can be observed. ANOVA revealed that these differences in topographies between initial samples are not statistically significant (for Sa: F = 1.02, *p* = 0.419; for S10z: F = 0.73, *p* = 0.556).

Differences in values of selected surface roughness parameters before and after the corrosion tests are negligible. Confidence intervals just slightly change after the tests, and in all investigated cases the bars of confidence intervals obtained before and after the corrosion tests overlap. This implies that the treatment did not cause statistically significant changes in tracked parameters. To confirm this, a series of paired *t* tests was performed, and the results are presented in Figure 3. Ho denotes the null hypothesis (mean of differences between two populations is zero), error bars denote 95% t-confidence intervals for the mean of differences, and $\bar{x}$ denotes the mean of differences. The data distribution was within the normality standard. For all samples, the null hypothesis could not be rejected since Ho was within the calculated t-confidence intervals. Therefore, the *t* test confirmed the initial observation that the media used was unable to cause a statistically significant change of selected surface roughness parameters for large-area scans.

### 3.2. Surface Roughness Parameters 10 × 10 µm

The average values of Sa and S10z parameters for areas of 10 × 10 µm determined before and after the corrosion tests are presented in Figure 4. These small sample areas are characterized by surface roughness parameters whose values belong to a nanometer range (Sa = 14–22 nm; S10z = 88–145 nm). This indicates very smooth surfaces with much lower surface roughness when compared to surfaces analyzed on the areas of 80 × 80 µm. Similarly, as was the case for 80 × 80 µm measurements, variations in the initial surface roughness parameters of approx. 25% also exist for the evaluation areas of 10 × 10 µm. The performed ANOVA analysis revealed that there is a significant difference between initial sample surfaces (for Sa: F = 3.94, *p* = 0.028; for S10z: F = 6.12, *p* = 0.006). The changes in Sa and S10z parameters induced by corrosion are not so pronounced. For these cases, the values of confidence intervals underwent only a minor change after the corrosion tests. Again, the bars of the confidence intervals before and after the corrosion tests overlap.

The developed interfacial area ratio (Sdr) and skewness (Ssk) parameters for the areas of 10 × 10 µm determined before and after the tests are presented in Figure 5. As was the case for other surface roughness parameters of the initial sample surfaces, it is also the case for the Sdr and Ssk parameters. These parameters vary between the samples up to approximately 40%. The performed ANOVA analysis showed a significant difference between the initial sample surface characterized for parameter Sdr (F = 3.77, *p* = 0.032) and a non-significant difference for parameter Ssk (F = 0.56, *p* = 0.649). The obtained values of the Ssk parameters indicate that the investigated surfaces do not have a pronounced polarity; they are mostly neutral. Corrosion tests induced some changes in Sdr and Ssk parameters, but these are not so pronounced. The values of the Sdr and Ssk confidence intervals mostly undergo a minor change after the corrosion tests. It is also noteworthy that the bars of confidence intervals in all investigated cases overlap.

The Anderson–Darling test confirmed that data fits the normal distribution, so a series of paired *t* tests was performed on measured roughness parameters determined for 10 × 10 µm areas. The results are presented in Figure 6 and Figure 7. The sample treated with artificial saliva (Sample 1) did not display significant changes in the studied surface roughness parameters. However, all samples treated with mouthwashes (Samples 2, 3, and 4) significantly altered the value of at least one surface roughness parameter (*p* < 0.05). Only the treatment for sample 4 did not cause a significant change in the Sa parameter (*p* = 0.232). Significant change in the Ssk parameter occurred only for treatment of Samples 2 (*p* = 0.024) and 4 (*p* = 0.027). Regarding the Sdr parameter, the treatment of Samples 2 (*p* = 0.049), 3 (*p* = 0.025), and 4 (*p* = 0.05) induced a significant change in this parameter, while treatment of sample 1 (*p* = 0.095) proved to be non-significant.

The paired *t* test showed that each treatment induced a statistically significant change in surface roughness parameters. For the development of valid clinical practice guides that assist practitioners, it would be beneficial to determine whether there is a difference between the mouthwashes used. To determine whether examined corrosion media cause identical or statistically significant change in surface roughness parameters, the ANOVA analysis was employed. The ANOVA revealed there was indeed a statistically significant difference between the examined media (for Sa: F = 9.02, *p* = 0.001; for S10z: F = 15.63, *p* = 0.001; for Sdr: F = 10.04, *p* = 0.001; for Ssk: F = 13.58, *p* = 0.001).

## 4. Discussion

### 4.1. Topography and Surface Roughness of Sample Areas of 80 × 80 µm

The initial topography of the investigated surfaces is dominated by micro-grooves, which is typical for elements produced by cold forming manufacturing processes without subsequent grinding or polishing. The differences of approx. 15% in the average values of samples initial surface roughness parameters (Sa and S10z) are a consequence of differences in quantity and depth of grooves in the evaluated areas of the samples’ surfaces.

The results of *t* tests revealed that none of the investigated corrosive media caused a significant change of the considered surface roughness parameters determined for the areas of 80 × 80 µm. The fact that changes in surface topography cannot be observed indicates either a very low intensity of corrosion or a complete absence of corrosion processes. These findings are not in line with the results from previously published studies [14,15,16,17,19,20,21,23,26]. In those investigations, a change in surface topography and decreased corrosion resistance of NiTi alloy treated with the investigated media was revealed. Such discrepancy could be a consequence of differences in testing conditions employed in the aforementioned publications. For example, the application of external potential and electrical currents in electrochemical testing could affect the corrosion processes [36,37,38]. However, the presence of corrosion of NiTi was also observed in non-electrochemical tests [30,31].

Moreover, we postulate additional reasons why the non-electrochemical corrosion tests from this study resulted in an insignificant change in the topography and surface roughness parameters. The aspiration to measure corrosion processes in realistic conditions led to the selection of commercially available archwire and its use in as-received condition. Relatively high values of Sa and S10z parameters of 80 × 80 µm areas, and their large scatter, are a consequence of uneven surfaces with deep grooves and high peaks. As such, the detection of smaller changes in roughness parameters that are caused by corrosion processes is hampered. This effect is so pronounced that the paired *t* test and one-way ANOVA failed to detect and compare nanometric changes that occurred in surface roughness parameters. Performing tens or even hundreds of measurements would undoubtedly increase statistical power to the point of being able to detect process differences. However, the use of this approach significantly increases the associated time and costs for analysis. Another, more efficient approach is to reduce variability by performing the analysis only on areas located between the grooves and/or high peaks. Examination of the performed measurements revealed that the largest area that fits these criteria for all samples is 10 × 10 µm. All measurements were cropped in this way and detailed statistical analysis was performed.

### 4.2. Topography and Surface Roughness of Sample Areas of 10 × 10 µm

Results of *t* tests indicate that the sample exposed to artificial saliva with pH 7.1 (Sample 1) exhibits an insignificant change in all roughness parameters. These findings do not correlate with previous investigations [23,26], where it is reported that artificial saliva induces the leaching of a relatively small amount of Ni ions [23], followed by changes on the surface (in electrochemical tests) [26].

The changes observed in topography on the areas of 10 × 10 µm for the samples treated with mouthwashes (Samples 2, 3, and 4) indicate that these corrosion processes induced surface changes on the nano-level. This finding confirms that the corrosion processes have a very low intensity whose effects can be statistically discerned only in the areas of low surface roughness. In the investigated cases these areas had been found only on surface segments of 10 × 10 µm.

Contrary to the initial large area measurements, ANOVA analysis revealed a statistically significant difference in the initial samples’ surface parameters. This result could be expected since sampling was not performed randomly, and the analysis was performed on very small areas (10 × 10 µm). A minor change in the values of confidence intervals, before and after the specific test, indicates uniform changes in nanotopography. The values of confidence intervals are mostly ~50% of the average values of the surface roughness parameters. This is quite a large value, which indicates a considerable variation in surface roughness parameters between different sample locations and a surface non-uniformity. Consequently, measurements made before the corrosion tests could not be grouped to perform a single ANOVA analysis, and changes in each sample had to be analyzed separately.

Results of *t* tests indicate that samples exposed to the investigated mouthwashes (Samples 2, 3, and 4) displayed a significant change in almost all concerned surface roughness parameters. Observed changes correlate with results obtained in investigations where it was shown that the presence of fluoride and chloride ingredients leads to a decrease in corrosion resistance [14,15,16,17,19,20,21], a change in surface [14,16,19,20,21], and an increase in the amount of Ni ions release from NiTi alloy [10,17,22,32,35]. However, the EDS analysis revealed that none of the investigated treatments caused a significant change in chemical compositions. This, coupled with the results of statistical analysis, suggests that mouthwashes in the investigated conditions induced corrosion of low intensity that is probably localized on thin surface layers only several nanometers in depth.

To reveal whether the corrosion process caused by specific media induces material loss, material gain, or both, it is necessary to determine the exact location of changes relative to the surface mean plane. By superimposing 3D topographic data before and after corrosion, one can calculate exact volumetric changes. Unfortunately, this kind of analysis could not be performed on the results of this study. All measured surfaces were subjected to corrosion and, therefore, one could not reliably determine the vertical axis and precisely superimpose one dataset over another. This could possibly be achieved by properly masking one part of the surface to completely preserve it and use it for matching pre- and post- corrosion measurements. This technique is not yet sufficiently refined. Therefore, in this study, we developed a methodology for assessing material gain or loss through the combined effect of various roughness parameters. The underlying logic behind the developed methodology is explained on an exemplary surface and graphically presented in Figure 8. The initial surface of the sample is approximated with a sinusoidally shaped profile with small peaks and valleys above and below the mean plane (Figure 8a). The possible effects of corrosion and the locations of their manifestation on the initial profile are schematically represented by symbols given in Figure 8b. After incorporating these transformations, a profile model is obtained. It contains all the changes that may occur on the initial profile (Figure 8c). By analyzing the formulas used to calculate them, the change in each surface roughness parameter (Sa, Sdr, Ssk) can be correlated with a specific dominant corrosion effect, i.e., material gain or loss relative to the mean profile plane. Through the elimination method, one can disqualify effects that cause opposite trends in surface roughness parameters and thus deduce a dominant effect of a specific corrosion medium (Sample) on surface topography.

The analysis is first performed for the change in the Sa parameter, as shown in Figure 9. The increase in this parameter, which is observed for Sample 2, could be caused by material gain above and/or material loss beneath the mean plane. All other possible effects would decrease the value of Sa. They could be present but can be excluded as non-dominant. Accordingly, a decrease in Sa values for Sample 3 can be explained by material loss above and/or material gain beneath the mean plane. Sample 4 did not exhibit a detectable change in the Sa parameter, which means that in this medium all corrosion effects can be present on the surface. The profile model for this sample looks like the exemplary surface shown in Figure 8c.

The value of the Sdr parameter for Sample 2 increased after corrosion experiments. Accordingly, effects that cause a reduction of the developed area can be excluded as non-dominant. Therefore, the simultaneous increase in Sa and Sdr values for Sample 2 can be explained if corrosion caused material gain above the mean plane and/or material loss beneath it. A similar analysis was performed for Samples 3 and 4, and the results are shown in Figure 10.

The analysis of corrosion effects that induce changes in the Ssk parameter is somewhat more complicated. It is well known that the high positive values of the Ssk parameter are an indication of dominant peaks above the mean plane, and vice versa [42]. However, this parameter is of high sensitivity and its changes cannot be easily linked with a certain corrosion effect on the surface topography. A significant change in Ssk could be caused by relatively low-intensity changes on peaks and valleys that are located quite distant from the mean plane. However, relatively high-intensity changes in the profile could cause repositioning of the mean plane relative to the unchanged features of the surface. Consequently, a vast number of unchanged peaks or valleys become more pronounced.

In order to depict these effects on the currently analyzed profiles, we performed simulated corrosion experiments where surface changes were known a priori. Thus, their effect on Ssk could be easily discerned. Two simulated corrosion attacks were carried out. Simulation 1 revealed the influence of the high-intensity corrosion effects, and simulation 2 revealed low-intensity corrosion effects on the change of parameter Ssk. For simplicity, the simulation of corrosion effects on the Ssk parameter is performed on a single profile taken from measurement performed before the corrosion experiment (Figure 11 and Figure 12). It is postulated that equivalent profile and surface (areal) roughness parameters have similar responses to similar topography changes. Simulation of a relatively high intensity localized corrosion attack on profile and its effect on parameters Ra and Rsk is displayed in Figure 11. The corrosion intensity is set in a way that changes on a profile cause a (relatively high) change of parameter Ra that is detected as significant in this investigation. After the simulated corrosion attack, the area above the mean plane decreased. This type of change causes the repositioning of the mean plane, downwards, relative to the unchanged part of the surface (Figure 11b). Consequently, it resulted in a decrease in the depth of the unchanged grooves and an increase in the height of the unchanged peaks. By comparing the values of the Rsk parameter of the initial profile and profile after simulated corrosion (without plane repositioning), it can be noticed that material loss caused a decrease in this parameter (Figure 11c). However, due to the repositioning of the mean plane (Figure 11c), it can be noticed that this process caused an increase in the Rsk parameter. Such changes occurred due to the fact that the unchanged peaks became higher, and the unchanged grooves became shallower.

Results of the simulated low-intensity corrosion attack on the peaks and grooves (simulation 2) and its effect on parameters Ra and Rsk are displayed in Figure 12. Low-intensity corrosion does not induce a notable change in Ra (Sa) and does not cause a pronounced repositioning of the mean plane. However, simulated material loss, i.e., increase in depth of grooves and decrease in height of peaks, leads to a decrease in Rsk values. It should be noted that parameter Rsk exhibits a major change only if changes in surface features occur at relatively long distances from the mean plane.

Simulated corrosion experiments proved that different intensities of the same corrosion effect could lead to both an increase and a decrease in the Rsk (Ssk). Large changes on the surface that are detectable by parameter Ra (Sa) cause a repositioning of the mean plane. Therefore, in case of a significant change of parameter Sa (Simulation 1), an increase in parameter Ssk could be interpreted as material loss, and vice versa. In contrast, an insignificant change in parameter Sa and a decrease in Ssk (Simulation 2) indicate a material loss, and vice versa.

For Sample 2, the values of both Sa and Ssk increased. Considering the previous analysis and discussion, the corrosion caused a material loss on the surface, and all other possible effects can be excluded as non-dominant. By performing a similar analysis, a dominant corrosion effect was determined for all investigated media, and the results are presented in Figure 13. Accordingly, the analysis revealed that treatments with fluoride-containing media, namely Aquafresh^®^ (Sample 2) and Listerine^®^ (Sample 4), cause material loss, while fluoride-containing Eludril^®^ (Sample 3) can cause both material loss above the mean plane and/or material gain beneath the mean plane.

Several investigations reported that media with chlorine-containing compounds causes material loss [10,24]. In the first few weeks of its contact with Ni-Ti alloy, due to the higher affinity of Cl ions toward Ni, it replaces the O in the protective oxide layer and induces Ni release [11,12]. This results in surface depassivation and consequently material loss [10,11,12]. However, published findings also indicate that a chlorine-containing compounds in the medium can induce the formation of corrosion products that remain on the surface [10,19]. Additionally, Hu et al. [13] reported that NiTi has the ability of self-healing, repassivation, which usually manifests in material gain. Therefore, the results presented herein correlate with both kinds of surface changes reported in the literature for NiTi corrosion in chloride-contained media, i.e., material loss [10,24] and material gain [10,11,12,13].

The mouthwashes with fluorine-containing compounds employed in this investigation contain a similar amount of fluoride. However, the apparent intensity of the corrosion and dominant locations of the corrosion attack were different. Both media caused material loss but induced opposite effects on surface roughness. Aquafresh^®^ (Sample 2) dominantly caused material loss beneath the mean plane which increased the surface roughness, while Listerine^®^ (Sample 4) induced material loss above and beneath the mean plane which consequently decreased surface roughness. This, together with the fact that commercial mouthwashes with fluorine-containing compounds do not cause the dissolution of Ti-O but a release of Ni-ions [43] suggests that the investigated mouthwashes behave in the same fashion. The opposite trends on surface roughness which is observed for fluoride media are in agreement with the literature data that fluoride compound-containing media can induce both an increase [30,31,32] and a decrease [33] in surface roughness. Our findings suggest that these diverging trends are a consequence of preferred corrosion locations, relative to the mean plane. Such behavior could be explained by a difference in wetting characteristics caused by the non-active ingredients of the mouthwash. Further research is required to obtain definitive mechanisms, especially when one takes into account studies that revealed that fluoride-containing media could also cause material gains [7,15]. Corrosion processes could be influenced by numerous additives and their influence deserves more careful consideration.

Results of ANOVA and *t* tests can be discussed from the standpoint of corrosion intensity, i.e., “aggressiveness”, between the investigated mouthwashes. The ANOVA comparisons revealed a significant difference between investigated mouthwashes. However, the absolute value of changes of almost all surface roughness parameters, except the Sa for Listerine ^®^, have the same order of magnitude. This finding suggests that the investigated mouthwashes have almost the same aggressiveness, although sometimes with different polarity, but a change in the Sa parameter, which is an indication of volume change, indicates that Listerine^®^ caused the lowest amount of corrosion. Although Listerine^®^ and Aquafresh^®^ have a similar fluoride concentration, Listerine^®^ induced significantly less corrosion, characterized by parameter Sa. This means that other ingredients in mouthwashes and/or their pH values also have a pronounced effect on corrosion processes. Additionally, it must be noticed that insignificant changes in the parameter Sa for the sample treated with Listerine^®^ could be caused by the inability of parameter Sa to detect the same type of surface changes that occurred above and beneath the mean plane simultaneously.

## 5. Conclusions

This investigation evaluated the topographic changes of NiTi alloy archwire exposed to artificial saliva and commercially available fluorine- and chlorine-containing mouthwashes in non-electrochemical corrosion tests. From the obtained results the following conclusions can be drawn:The employed experimental setup achieved highly accurate lateral positioning of AFM measurements before and after corrosion tests. The positioning error was approx. 0.1 µm, and it enabled the detection of surface changes induced by low-intensity (realistic) corrosion processes on predefined locations.Due to high initial surface roughness and low-intensity corrosion, AFM and SEM analyses performed on large areas (80 × 80 µm) were not able to detect surface changes caused by studied mouthwashes. However, the exclusion of large grooves revealed statistically significant nanotopographic changes in the studied surfaces. Considering that EDS analysis did not reveal any changes in surface chemical composition, it is suggested that corrosion processes induce changes localized on thin surface layers only several nanometers in depth.The sample exposed to artificial saliva did not display statistically significant changes in any surface roughness parameter.A novel analysis methodology was developed to obtain insight into locations of material gain or material loss based on standard surface roughness parameters Sa, Sdr, Ssk, and S10z. The developed methodology revealed that mouthwashes (Aquafresh^®^ and Listerine^®^) with fluorine-containing compounds dominantly cause material loss, while mouthwash (Eludril^®^) with chlorine-containing compounds can cause both material loss and material gain.Both fluoride compound-containing mouthwashes caused material loss but induced opposite effects on surface roughness. Findings suggest that these diverging trends are a consequence of preferred corrosion locations, relative to the mean plane. Such behavior could be explained by a difference in wetting and corrosion characteristics caused by the non-active ingredients of the mouthwash. Further research is required to obtain definitive mechanisms.

## Figures and Tables

**Figure 1 materials-15-08705-f001:**
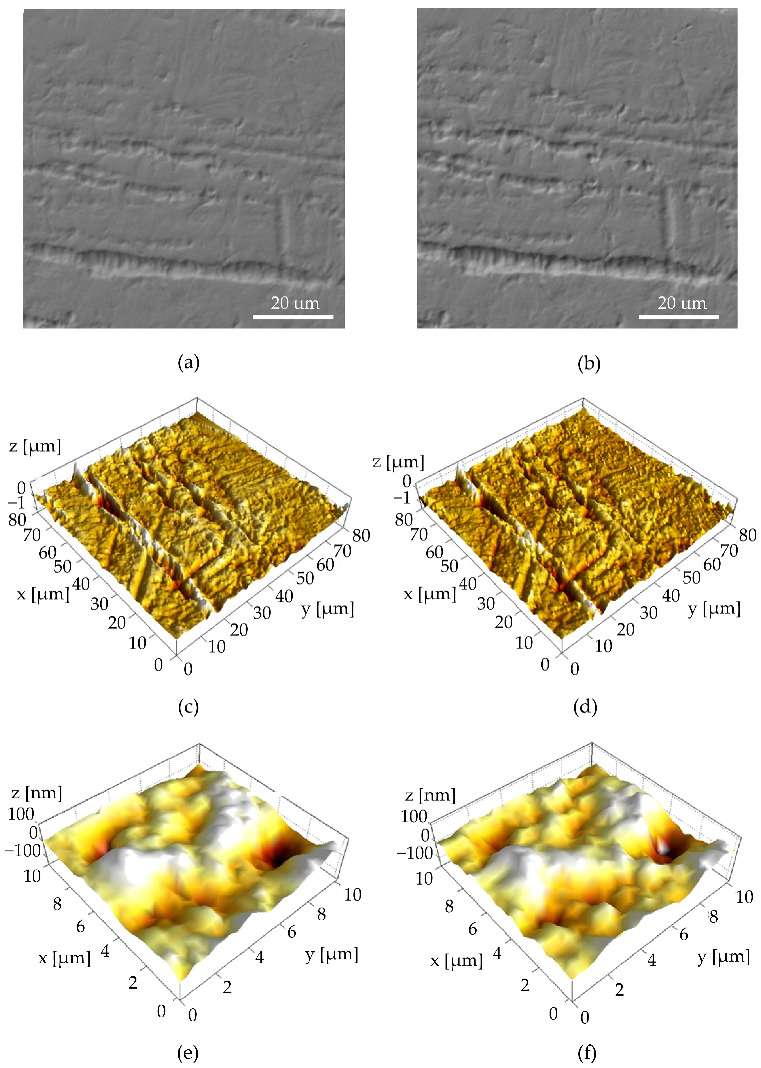
Representative images before (left) and after the corrosion test (right) in Eludril CLASSIC^®^ (Sample 3): (**a**,**b**) SEM images of 80 × 80 µm area; (**c**,**d**) AFM topography images of the same 80 × 80 µm area; (**e**,**f**) AFM topography images of cropped 10 × 10 µm smooth area between the grooves.

**Figure 2 materials-15-08705-f002:**
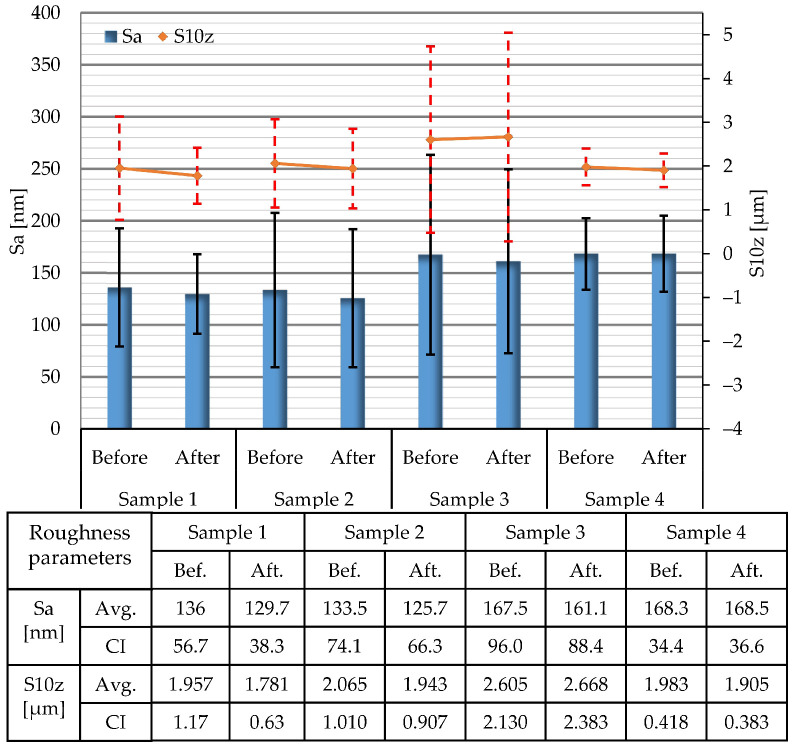
Average values (Avg.) of Sa and S10z surface roughness parameters determined on areas of 80 × 80 µm and corresponding confidence intervals (CI).

**Figure 3 materials-15-08705-f003:**
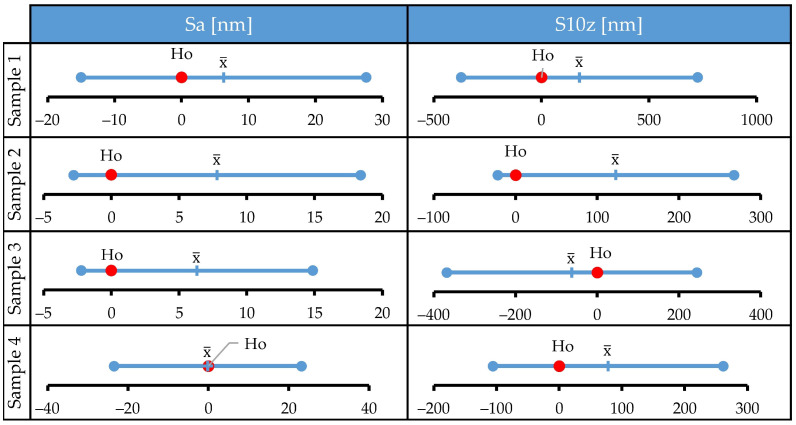
Results of a paired *t* test: statistical analysis of the changes in Sa and S10z parameters determined on areas of 80 × 80 µm before and after the corrosion tests.

**Figure 4 materials-15-08705-f004:**
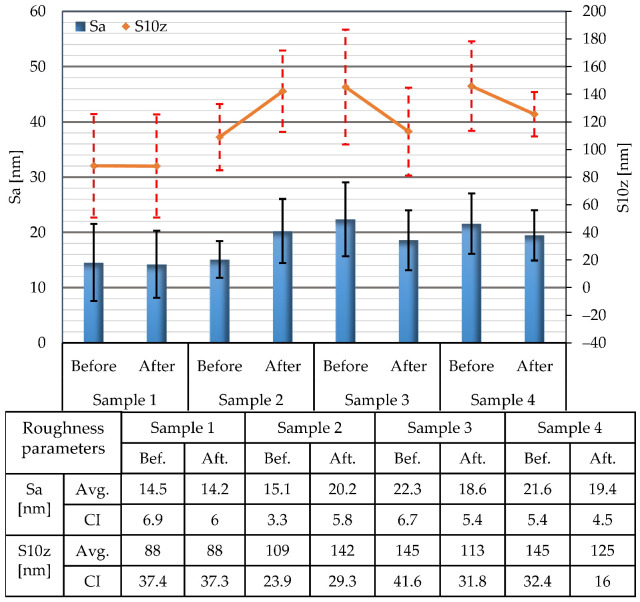
Average values of Sa and S10z surface roughness parameters determined on areas of 10 × 10 µm and corresponding confidence intervals (CI).

**Figure 5 materials-15-08705-f005:**
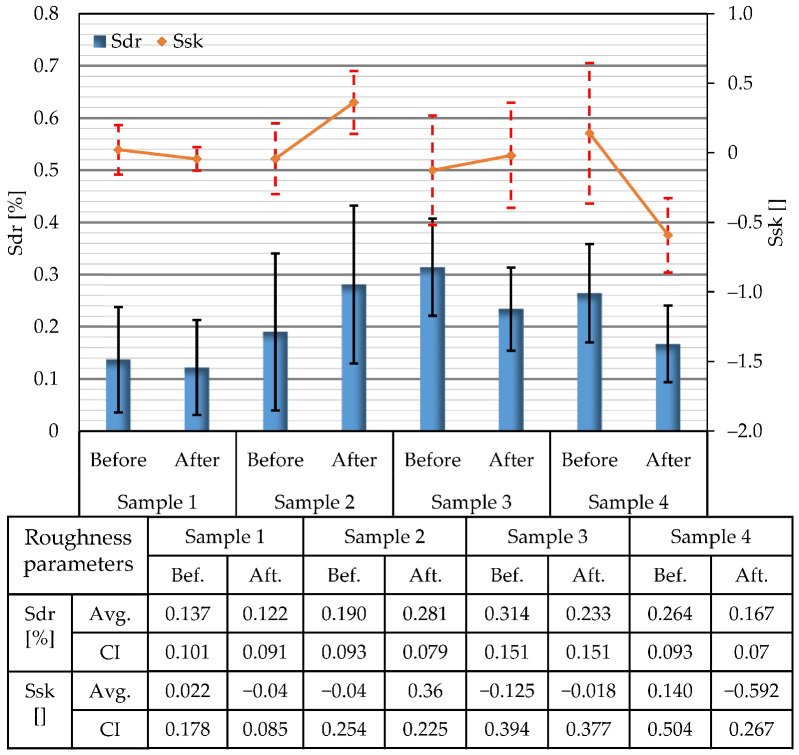
Average values of Sdr and Ssk surface roughness parameters determined on areas of 10 × 10 µm and corresponding confidence intervals (CI) of all investigated samples.

**Figure 6 materials-15-08705-f006:**
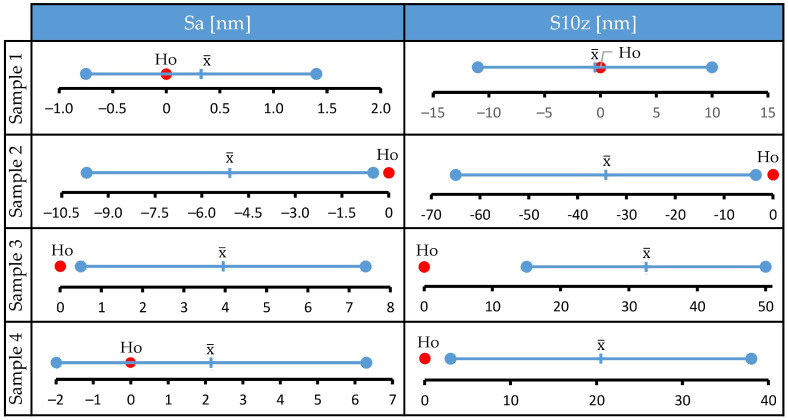
Results of a paired *t* test: statistical analysis of the changes in Sa and S10z parameters determined on areas of 10 × 10 µm before and after the corrosion tests of all samples.

**Figure 7 materials-15-08705-f007:**
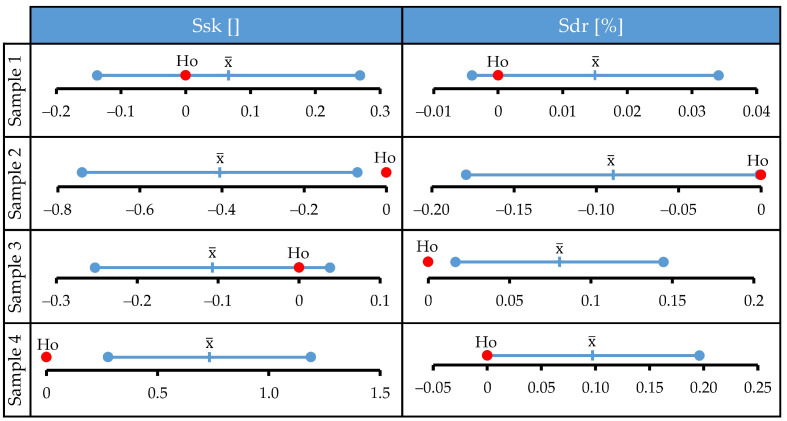
Results of a paired *t* test: statistical analysis of the changes in Ssk and Sdr parameters determined on areas of 10 × 10 µm before and after the corrosion tests of all samples.

**Figure 8 materials-15-08705-f008:**
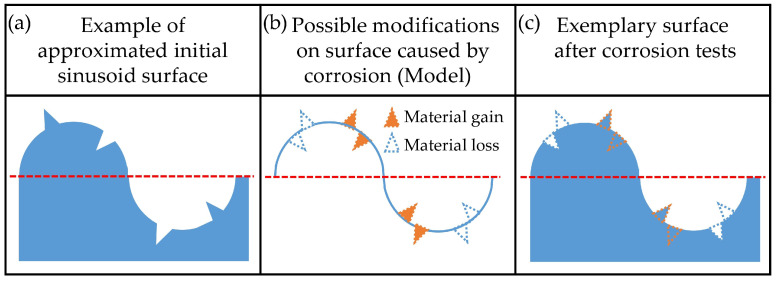
Graphical presentation of the analysis performed for detection of corrosion effects on surface topography: (**a**) approximated initial profile (before treatment); (**b**) possible corrosion effects and their locations; (**c**) profile of exemplary surface after corrosion.

**Figure 9 materials-15-08705-f009:**
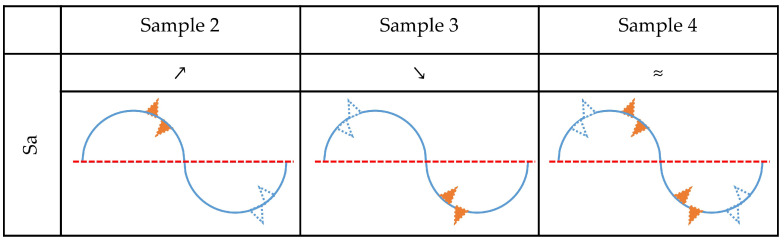
Determination of dominant corrosion effects that correlate with changes of parameter Sa. The empty blue triangle represents material loss while the orange filled triangle represents material gain. Results of paired *t* test are symbolically represented by symbols ≈, ↗, ↘. These symbols represent insignificant difference, significant increase, and significant decrease in specific parameters, respectively.

**Figure 10 materials-15-08705-f010:**
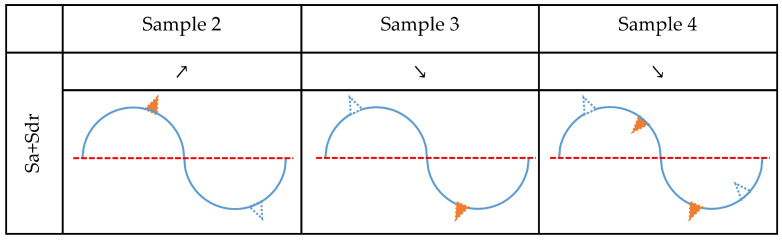
Determination of dominant corrosion effects that correlate with changes of parameter Sa and Sdr. The empty blue triangle represents material loss while the orange filled triangle represents material gain. Results of paired *t* test are symbolically represented by symbols ≈, ↗, ↘. These symbols represent insignificant difference, significant increase, and significant decrease in specific parameters, respectively.

**Figure 11 materials-15-08705-f011:**
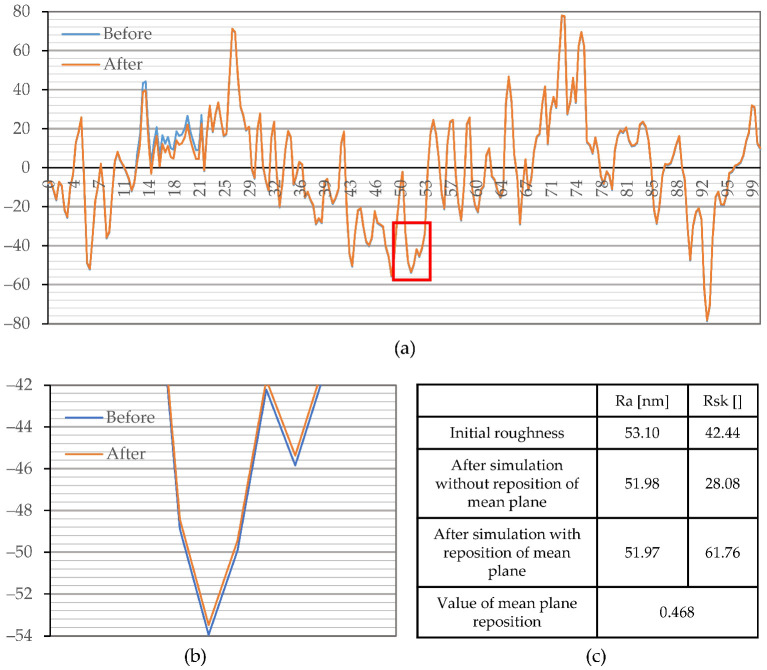
Simulation of localized and high-intensity corrosion attack (Simulation 1): (**a**) profiles employed for analysis; (**b**) magnified marked location, on the profile employed for analysis, that depicts relative size of mean plane repositioning; (**c**) change in tracked parameters.

**Figure 12 materials-15-08705-f012:**
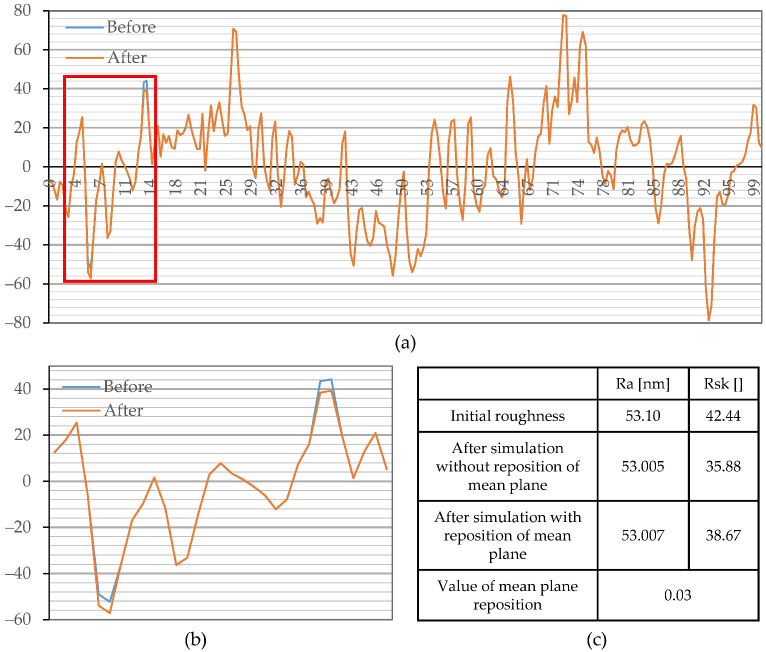
Simulation of localized and low-intensity corrosion attack (Simulation 2): (**a**) profile employed for analysis; (**b**) magnified marked location, on profile employed for analysis, that depict simulated effect of corrosion; (**c**) change in tracked parameters.

**Figure 13 materials-15-08705-f013:**
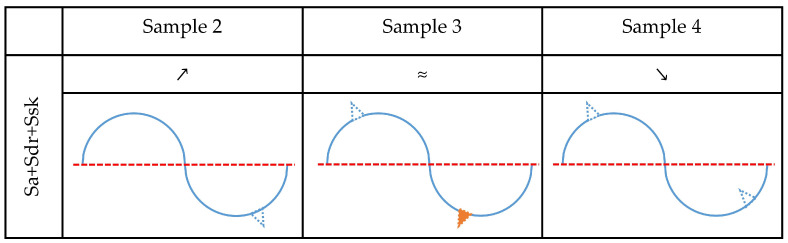
Dominant corrosion effects that correlate with changes in parameters Sa, Sdr, and Ssk. The empty blue triangle represents material loss while the orange filled triangle represents material gain. Results of paired *t* test are symbolically represented by symbols ≈, ↗, ↘. These symbols represent insignificant difference, significant increase, and significant decrease in specific parameters, respectively.

**Table 1 materials-15-08705-t001:** Sample denotations, used media, and their main corrosive ingredients.

Sample	Medium	Main Corrosive Ingredients
Sample 1	Artificial saliva (Pharmacy “Belgrade”)	-
Sample 2	Aquafresh Big teeth^®^ (GSK Consumer Healthcare)	Fluoride-containing compounds 0.05% (255 ppm fluoride)
Sample 3	Eludril CLASSIC^®^ (Pierre Fabre medicament)	Chlorine-containing compounds 0.6%
Sample 4	Listerine^®^ (Green Tea) (Johnson & Johnson)	Fluoride-containing compounds 0.05% (220 ppm fluoride)

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
