# Peer review of "Nanotopography Evaluation of NiTi Alloy Exposed to Artificial Saliva and Different Mouthwashes"

_materials, 2022, doi:10.3390/ma15238705_

Round 1

Reviewer 1 Report

I would like to congratulate the authors as this work was carried out and presented in an impeccable way.  Every section is well developed, supported and explained.  However, I would suggest the authors to include some information that was missing in the Materials and Methods section.  The number of samples per treatment and the storage conditions (other than room temperature) are not there.  Whether the mouthwashes and artificial saliva were changed, and if so, how often, during the testing period would be interesting to know because the exposure time is high for a continuos exposure to a mouthwash and conditions might change, especially for the mouthwashes after their interaction with the NiTi wires.

Reviewer 2 Report

 The article «Nano topography evaluation of NiTi alloy exposed to artificial saliva and different mouthwashes» is devoted to a relevant topic and is of interest to the reader. However, the article must be subjected to a major revision.

 The introduction is written in sufficient detail, but it is necessary to clearly formulate the purpose of the study. In the first sentence, it is necessary to clarify that we are not talking about weight, but about atomic percentages.

The Materials and Methods section does not contain information on the number of samples for each study. The reviewer is not sure about the correctness of the indication of the trade names of liquids. It is not clear why such a research time was chosen - 21.5 days.You need to add subsections to the results part

The authors incorrectly indicated the chemical composition of nickelide titanium. 

«The EDS analysis revealed that the chemical composition of the sample has not significantly changed after exposure to the corrosive media and was comprised of approximately 40 wt%Ti and 60 wt%Ni.»

For a near-equiatomic alloy, there must be - 54,85-54,85 wt%Ni or 50,0-50,25 at%Ni

It is not certain that the authors carried out this study.

Please describe ANOVA and T-test in more detail.

Results should include more SEM and AFM topography images

It would be useful to see if there are changes in the structure using XRD and TEM methods. This would allow new results to be added for discussion.

Conclusions formulated acceptable

The research carried out in the article in its present form is not sufficient for publication in a high-ranking journal.

It seems to me that the use of commercial names for mouthwashes is not acceptable and is advertising.

Structural studies must be added to publish an article.

Round 2

Reviewer 2 Report

The authors made changes to the text of the article and took into account most of the comments of the reviewer. I think that in its current form the article can be published